# Optimizing Sampling Technique Parameters for Increased Precision and Practicality in Annual Bluegrass Weevil Population Monitoring

**DOI:** 10.3390/insects14060509

**Published:** 2023-05-31

**Authors:** Ana Luiza Viana de Sousa, Olga S. Kostromytska, Shaohui Wu, Albrecht M. Koppenhöfer

**Affiliations:** Department of Entomology, Rutgers University, 96 Lipman Dr., New Brunswick, NJ 08901, USA; sousa.alvs@gmail.com (A.L.V.d.S.); okostromytsk@umass.edu (O.S.K.); wu.6229@osu.edu (S.W.)

**Keywords:** annual bluegrass weevil, extraction, sampling, monitoring, turfgrass

## Abstract

**Simple Summary:**

The annual bluegrass weevil (ABW) is the most severe insect pest of short-mown turfgrass in eastern North America, and the proper monitoring of overwintered adult populations in spring is critical to the timely management of this significant pest. We evaluated three sampling methods including soap flushing, vacuuming, and mowing in golf course putting greens and fairways for monitoring ABW adults. Soap flushing was the most efficient method, and adult extraction efficiency was not affected by the temperature or time of day; both vacuuming and mowing were more efficient for adult recovery on greens vs. fairways; the efficiency of mowing was affected by the time of day and temperature, but vacuuming efficiency was not. Hence, soap flushing was determined to be the optimal method for monitoring adult ABWs, while vacuuming may serve as an alternative sampling method on greens.

**Abstract:**

The annual bluegrass weevil (ABW), *Listronotus maculicollis* (Kirby), a significant pest of short-mown turfgrass in eastern North America, has developed widespread insecticide resistance because of excessive synthetic insecticide use. The proper monitoring of this pest may reduce insecticide applications in time and space. This study evaluated three sampling methods (soap flushing, vacuuming, and mowing) in golf course greens and fairways for monitoring adult ABW. Soap flushing was the most efficient method, especially with an 0.8% solution in two portions of 500 mL, extracting over 75% of the adults, and the extraction efficiency was not affected by the temperature or time of day. Vacuuming was more effective for recovering adult ABWs on greens (4–29% extracted) than on fairways (2–4%) but was not affected by the time of day. The extraction of adult ABWs in mower clippings was significantly affected by mowing height (higher recovery from greens versus fairways), and the efficiency decreased with the temperature. Adding a brush to the mower increased adult removal (from 15% to 24%) in greens at higher temperatures (18–25 °C); 70% of adults recovered in the clippings were unharmed. Overall, our findings suggest that soap flushing should be the preferred method for monitoring adult ABWs, and vacuuming might be a viable alternative for greens.

## 1. Introduction

The larvae of the annual bluegrass weevil (ABW), *Listronotus maculicollis* (Kirby), cause extensive damage to high-profile turfgrass areas (greens, collars, approaches, fairways, tee boxes) on golf courses in eastern North America [1]. The first three larval instars, feeding inside grass stems, cause limited damage, whereas the last two instars cause severe damage by feeding on crowns and the base of stems from shallow burrows in the thatch or soil. Pupation occurs in the soil/thatch near the soil surface. Adults are active within the turfgrass canopy where they feed and mate. ABW adults overwinter in tall grass or under tree litter near infested playing surfaces on golf courses [2]. Overwintered adults migrate back to short-mown turfgrass areas in early spring where they feed, mate, and oviposit in the plant stem over the course of several weeks [1,3].

ABW is a difficult-to-control pest with 2–3 generations per year, multiple life stages present simultaneously, and increasing asynchrony in their stages during the growing season. Golf course superintendents have relied primarily on synthetic insecticides for ABW management [4], but excessive insecticide use has led to widespread insecticide resistance [4], first recognized in pyrethroids [5] but later also in insecticides from several other classes [6,7,8]. Against highly resistant ABW populations, adulticides (pyrethroids, chlorpyrifos) provide no control, and several larvicides (e.g., chlorantraniliprole, clothianidin, indoxacarb, trichlorfon) are also affected by resistance [6,7,8,9]. Golf course superintendents who do not use adulticides in spring prefer to target the younger larval stages of the spring generation with systemic insecticides so that they have time for a follow-up application against the mid-size larvae if necessary [4].

In attempts to mitigate and delay resistance development in ABW populations, insecticides have to be applied only when and where necessary. That, however, requires monitoring and sampling methods with high predictive power that are easy to use and fit into superintendents’ busy schedules. The damage threshold for significant turf damage is estimated at 30–80 ABW larvae per 0.093 m^2^ (square foot) [1,10] but may vary considerably with temperature, soil moisture, mowing height, grass species composition, and potentially other factors. Adult densities requiring control measures are even less understood since the adults are not the damaging stage, and larval densities arising from a given adult density could be affected by all the above factors. Based on many years of observations, adult densities of 10 per 0.093 m^2^ and above are likely to result in potentially damaging larval densities (AMK; personal observations).

Several methods are currently available to monitor adults or larvae [1,10]. Larval monitoring has greater predictive power since larvae are the stage that causes significant damage, and adult densities only indirectly predict larval densities. However, larval sampling methods (submersion of turf cores in salt water or heat extraction of cores) are generally more disruptive to the turf, more time-consuming, and harder to implement in the field without additional equipment and expertise than adult monitoring methods. Larval sampling methods are widely used by scientists and consultants but are not likely to be used by superintendents.

The monitoring methods most likely to be used by superintendents involve sampling adult ABWs via vacuuming, soap flushing, or clipping examinations [1]. Vacuuming using a leaf blower with inverted air flow is the only method that has been explored regarding the prediction of larval populations. McGraw and Koppenhöfer [11] developed binomial sequential sampling plans for forecasting larval populations based on adult counts. The method was developed on fairway turf and involves repeated vacuuming of standard-size samples (the length of the vacuuming period) during the spring migration of overwintered ABW adults until they are past peak densities. However, the adult ABW extraction efficiency of the method was not determined, and various factors are likely to influence the efficiency of vacuuming. We suspected that the efficiency tends to be higher at lower mowing heights and at higher temperatures when adult ABWs are more likely to be active on surfaces.

Adult ABWs are also known to be picked up by mowers and can be quantified in the mower clippings. We suspected that the percentage of adults picked up in mower clippings will also be higher with shorter mowing heights and when it is warmer. Czyzewski and McGraw [12]) observed in greenhouse pot experiments under constant and ideal conditions that the percentage of adults picked up by mowing was inversely related to mowing height, reaching 26–38% at the lowest height (2.5 mm), but these observations need to be confirmed in the field.

Soap flushing is the adult ABW extraction method least likely to be affected by environmental conditions and mowing height. Water mixed with liquid dishwashing detergent is applied to a specified area to irritate the adults to move to the surface and climb up the grass blades where they can be counted. Soap flushing is widely recommended for monitoring various surface-active turfgrass insects, but the recommended detergent concentration and water volume applied per sample area vary [1,10]. Optimal combinations are likely to differ between insect species, and extraction efficiency and any effects of environmental factors thereon have generally not been examined.

The goal of this research was to optimize the use and predictive power of sampling/monitoring methods for ABW larvae and adults so that they could be readily adopted by superintendents for the timely control of the pest problem. Specifically, we wanted to determine the effect of temperature and mowing height on the percentage of adults detected (1) in mower clippings, (2) via vacuuming with a leaf blower, and (3) via soap flushing. In addition, (4) we wanted to determine the effect of the soap solution volume applied and the detergent concentration on the extraction efficiency of the soap flushing method.

## 2. Materials and Methods

### 2.1. Insects and General Methodology

ABW adults to be released in mark-and-recapture experiments (Section 2.2 and Section 2.3) were collected by hand and via vacuuming from an infested green at Rutgers Horticultural Farm No. 2 (North Brunswick, NJ, USA). Vacuuming was performed with a leaf blower (Echo ES-250 Shred “N’ Vac, ECHO Inc., Lake Zurich, IL, USA) with inverted air flow and fitted with a mesh insert to capture adults and debris. The adults were kept for no more than 3 days in containers on moist sand and fed with *Poa annua* L. clippings and a black cutworm, *Agrotis ipsilon* (Hufnagel), diet (Bio-Serv, Frenchtown, NJ, USA). For marking, the adults were placed in containers with a mixture of sand and color marking powder (DayGlo A-19, DayGlow Color Corp., Cleveland, OH, USA) and shaken for 30 s. We had previously determined over a 48 h observation period that this color powder was safe for the adults, did not affect their behavior and survival, and stayed on the adults long enough for the purposes of these experiments.

All experiments were conducted in turfgrass fields at Rutgers Horticulture Farm No. 2 that were maintained by the maintenance staff of the farm using standard conditions for golf course putting greens or fairways, except that insecticides were only applied, if necessary, after the experiments were concluded in each year. These fields are henceforth referred to as greens and fairways. Fairways were mown three times per week at a 9.5 mm height, which represents the lower end of typical fairway mowing heights. Greens were mown daily at 3.2 mm, which represents the higher end of typical putting green mowing heights.

### 2.2. Effect of Mowing Height and Turf Brush on Recovery of Color-Marked Adults in Mower Clippings

The experiment was conducted in a green and a fairway, which both consisted of a mix of approximately 25% *P. annua* and 75% creeping bentgrass, *Agrostis stolonifera* L., and had very low densities of a natural ABW population, as determined with soap extraction using the same method as described below in this section. In both fields, six plots of 183 cm length × 61 cm width arranged in parallel were flagged out for each of the two experimental runs. Forty color-marked ABW adults were released into a 30.5 cm long and 10 cm wide area in the center of the plot (i.e., at least 76.2 cm and 25.5 cm from the plot borders length- and widthwise, respectively) and allowed to disperse for 40 min. Then, the plots were mown individually with a walk-behind mower (Toro Greensmaster^®^ FlexTM; 53 cm cutting width; The Toro Company, Bloomington, MN, USA), making one pass down the long side of the plot. Plots were all mown in the same direction. The grass clippings were collected in a separate plastic bag for each plot. Half of the plots were mown without a brush attached; the other half were mown with a brush pad (TurfTrainerTM, North Scituate, MA, USA) attached to the front of the mower basket so that the brush brushed the grass in front of the mower blades. Such brushes are used to improve the consistency and smoothness of short-mown playing surfaces on golf courses. We hypothesized that the brush would either activate any adult ABWs on the turf surface or dislodge them if partially submerged between the turfgrass plants.

Directly after mowing, we determined the number of adult ABWs remaining in the plots by irritating them to the turf surface with an irritant soap flush and transferring them with soft forceps into 30 mL plastic cups. Two liters of liquid dishwashing detergent (0.4% Joy Ultra lemon-scent, Procter & Gamble, Cincinnati, OH, USA) solution was distributed over the center 61 cm × 61 cm of each plot at the beginning of a 15 min observation period and again 5 min after the first application. Adults were collected after 5, 10, and 15 min. Adults were collected from within that defined area only, as with all the following experiments. The cups with the adults and the plastic bags with mower clippings were brought to the laboratory for evaluation. The adults were categorized under a dissection microscope at 10× magnification as fully intact, alive with minor damage, and dead with or without damage. The first experimental run was conducted on 24 May 2018 (900–1000 h, sunny, 18–21 °C), and the second run was on 25 May 2018 (900–1000 h, sunny, 23–25 °C). For both runs, the last mowing of the experimental areas occurred about 24 h (green) and 48 h (fairway) before the experimental mowing.

### 2.3. Effect of Mowing Height on Recovery of Color-Marked Adults through Turf Vacuuming

The experiment was conducted in the same green and fairway as the experiment on recovery in mower clippings. Plots consisted of twelve 61 cm × 61 cm areas arranged in three rows with a 61 cm buffer within and between rows. Forty color-marked ABW adults were released in the center 10 cm × 10 cm of each plot and allowed to disperse for 40 min. Adults were vacuumed up using an inverted leaf blower, described above. The blower was set to full power and dragged across the turf at a speed of approximately 3 km/h with the opening of the intake tube tight on the turf surface. Vacuuming treatments consisted of (1) no vacuuming, (2) vacuuming the entire plot for 15 sec, and (3) vacuuming the entire plot twice for a total of 30 sec. There were four replicates per treatment in both experimental runs. Vacuuming duration was based on previous observations that vacuuming an area of 61 cm × 61 cm using our normal method for monitoring ABWs took about 15 s.

Directly after vacuuming, we determined the number of adult ABWs remaining in the plots with a soap irritant applied to the entire plot using the same method as described in the experiment on adult recovery from mower clippings, except that adults were not examined for injuries. The first experimental run was conducted on 24 May 2018 (1000–1100 h, sunny, 21–23 °C), and the second experiment was on 25 May 2018 (1000–1100 h, sunny, 25–27 °C).

### 2.4. Effect of Water Volume and Detergent Concentration on ABW Adult Extraction Efficiency of Soap Flushing

Experiments examined the effect of application frequency, application volume, detergent (Joy Ultra lemon-scent) concentration, and sampling intervals on the extraction efficiency of the soap dilution for adult ABWs. They were conducted in a fairway consisting of a mixture of 50–70% *P. annua* and 30–50% *A. stolonifera* with a natural infestation of ABWs. Plots (30.5 cm × 30.5 cm) were arranged in a randomized complete block design with a 30.5 cm buffer between plots within rows and 91.5 cm between rows. Soap solution was carefully applied from a beaker throughout the plots so that no solution would run outside the plot. Emerged ABW adults were collected with soft forceps from the turf canopy at 5, 10, 15, and 20 min after the first soap solution application. For each time interval and each plot, adults were collected into separate 30 mL plastic cups and counted in the laboratory. For each experimental run, a new area within the same field was used.

In 2019, treatments were (1) two applications (at 0 and 5 min of the observation period) of 500 mL soap solution at 0.2% detergent; (2) two applications (at 0 and 5 min) of 500 mL soap solution at 0.4% detergent; (3) two applications (at 0 and 5 min) of 500 mL soap solution at 0.8% detergent; (4) one application (at 0 min) of 500 mL soap solution at 0.4% detergent; and (5) one application (at 0 min) of 1000 mL soap solution at 0.4% detergent (see also Figure 3). The experiment was run 4 times with a total of 19 replicates (22 April, 1000–1100 h, 13–15 °C, 6 replicates; 23 April, 930–1030 h, 15–17 °C, 3 replicates; 29 April, 1030–1130 h, 10–12 °C, 4 replicates; 6 May, 1030–1130 h, 14–15 °C, 6 replicates).

In 2020, treatments were (1) two applications (at 0 and 5 min) of 500 mL soap solution at 0.4% detergent; (2) two applications (at 0 and 5 min) of 500 mL soap solution at 0.8% detergent; (3) one application (at 0 min) of 500 mL soap solution at 0.4% detergent; (4) one application (at 0 min) of 500 mL soap solution at 0.8% detergent; and (5) one application (at 0 min) of 1000 mL soap solution at 0.4% detergent (see also Figure 3). The experiment was run twice with a total of 15 replicates (4 April, 1000–1100 h, 8–10 °C, 6 replicates; 7 April, 930–1030 h, 12–14 °C, 9 replicates).

### 2.5. Effect of Time of Day and Mowing Height on ABW Adult Recovery in Mower Clippings

The experiment was conducted on a green with a mixture of approximately 25% *P. annua* and 75% *A. stolonifera* with natural infestations of ABWs. Twelve plots of 183 cm × 61 cm arranged in parallel were flagged out for each experimental run. Then, the plots were mown individually, clippings were collected, and any remaining adult ABWs were extracted with the soapy solution using the same methodology as described above for the experiment with color-marked adult ABWs, except that mowing was always performed without a brush attachment and soap extraction was performed over 20 min (500 mL of 0.8% soap solution at 0 and 5 min; adults collected at 5, 10, 15, 20 min). Experimental runs were conducted when the temperature was expected to be around 12 °C in the morning (around 900 h) and around 20 °C in the early afternoon (around 1330 h). Half of the plots were mown in the morning and the other half in the afternoon with treatments arranged so that plots with different mowing times alternated to minimize any potential effect of uneven adult ABW distribution. A total of six experimental runs were conducted on 4 May 2019 (15.0 °C and 21.7 °C), 6 May 2019 (6.7 °C and 18.3 °C), 29 April 2020 (7.8 °C and 19.4 °C), 28 April 2022 (10.0 °C and 21.1 °C), 2 May 2022 (12.2 °C and 22.2 °C), and 4 May 2022 (13.3 °C and 26.1 °C).

### 2.6. Effect of Time of Day and Mowing Height on ABW Adult Recovery via Vacuuming and Soap Flushing

The experiment was conducted on a green with a mixture of approximately 25% *P. annua* and 75% *A. stolonifera* and a fairway consisting of a mixture of 60% *P. annua* and 40% *A. stolonifera*, both fields with natural ABW infestations. Plots (61 cm × 61 cm) were arranged in a randomized completed block design with eight replicates per treatment in two rows without spacing. Treatments were (1) no vacuuming and (2) vacuuming the entire plot for 15 sec. Adults collected with the vacuum were placed in 30 mL plastic cups for evaluation in the laboratory. After vacuuming, any remaining adults were extracted in a central 30.5 cm × 30.5 cm area in each plot with the soapy solution (500 mL of 0.8% soap solution at 0 and 5 min; adults collected at 5, 10, 15, 20 min) and collected in 30 mL plastic cups for evaluation in the laboratory. The experiment was conducted when the temperature was expected to be around 12 °C in the morning (around 900 h) and around 20 °C in the early afternoon (around 1330 h). Two experimental runs were conducted in 2020 on 29 April (12.2 °C and 20.6 °C) and 30 April (15.0 °C and 20.0 °C).

### 2.7. Statistical Analysis

In all experiments, data analyzed were the total number of adults collected by the specific extraction method examined. Except in the experiment examining soap flushing (Section 2.4 and Section 3.3), we also analyzed the total number of adult weevils recovered by the examined extraction method plus the ensuing soap flush and the percentage of adults recovered in the examined extraction method relative to the total recovered with the examined extraction method plus the ensuing soap flush. To calculate the percent recovery of adult ABWs in the clippings and the vacuum samples in the experiments where the entire plot was not soap extracted, the number of soap-extracted adults was multiplied by the factor of “total area mown (or vacuumed)/total area soap-extracted” (2.1 and 4.0 in the experiments on the effect of temperature on the recovery of adult ABWs in mower clippings and vacuum samples, respectively). Data in all experiments were normalized as necessary with the square root, logarithm, or arcsine of the square root transformation and subjected to factorial analysis of variance (software: Statistix 10.0; Analytical Software, Tallahassee, FL, USA) with the experimental run, mowing height, time of day (morning vs. early afternoon), and extraction type as factors, as appropriate for each experiment (see Results section). For the experiment on the effect of time of day on the extraction of adults in mower clippings, temperature was included as a co-factor to time of day in the analysis, as there was a significant range of temperatures both in the morning and afternoon sampling events. In the experiment on the effect of time of day on extraction via vacuuming, there were fewer sampling events with a much smaller range of temperatures, so in this experiment, temperature was not included as a co-factor. Means were separated using Tukey’s HSD test (α = 0.05). Means presented in the text are followed by ±SEM (standard error of the mean).

## 3. Results

### 3.1. Effect of Mowing Height and Turf Brush on Recovery of Color-Marked Adults in Mower Clippings

The number of adults recovered in the clippings was significantly affected by the mowing height (F = 70.01; df = 1, 23; *p* < 0.001) and the presence of a brush (F = 4.58; df = 1, 23; *p* < 0.05), but both factors interacted significantly (F = 4.58; df = 1, 23; *p* < 0.05) (Figure 1). However, the total recovery of adults (combined from clippings and soap flushing) was almost twice as high (F = 51.76; df = 1, 23; *p* < 0.001) from the fairway (33.2 ± 1.4 without a brush; 32.2 ± 1.9 with a brush) compared with the green (14.7 ± 1.8 without a brush; 14.0 ± 1.6 with a brush). The brush had no effect on the total recovery (F = 0.29; df = 1, 23; *p* = 0.60), and there was no interaction between the mowing height and brush (F = 0.04; df = 1, 23; *p* = 0.84) (Figure 1).

To adjust for differences in the recovery rate between the fairway and the green, the data were converted into a percentage recovery (adults in clippings divided by adults total). The percentage recovery in the clippings was significantly affected by mowing height (F = 77.13; df = 1, 23; *p* < 0.001) and the presence of a brush (F = 4.58; df = 1, 23; *p* < 0.05), but both factors interacted significantly (F = 4.58; df = 1, 23; *p* < 0.05). The percentage recovery in the clippings was the lowest from the fairway with and without a brush (0.5 ± 0.5%), significantly higher from the green without a brush (15 ± 2%), and the highest from the green with a brush (24 ± 4%). Of the adults recovered from the green clippings, without (in parentheses, with) a brush, 69% (70%) were alive and undamaged, 13% (19%) were alive with minor damage, and 19% (11%) were dead.

### 3.2. Effect of Mowing Height on Recovery of Color-Marked Adults by Vacuuming

The number of adults recovered by vacuuming was about three times higher from the green than from the fairway (F = 40.80; df = 1, 31; *p* < 0.001) without a significant effect from the vacuuming duration (F = 0.13; df = 1, 31; *p* = 0.72) and no interaction between the mowing height and vacuuming duration (F = 0.36; df = 1, 31; *p* = 0.55) (Figure 2). However, the total recovery of adults (combined from vacuuming and soap flushing) was about twice as high (F = 300.89; df = 1, 47; *p* < 0.001) compared with the fairway (33.8 ± 1.4 with 15 s; 30.3 ± 1.3 with 30 s; 33.0 ± 0.9 with the soap flush only) as from the green (15.5 ± 1.1 with 15 s; 15.3 ± 0.8 with 30 s; 16.9 ± 1.4 with the soap flush only) without a significant effect from the vacuuming duration (F = 1.73; df = 2, 47; *p* = 0.19) and no interaction between the mowing height and vacuuming duration (F = 0.76; df = 1, 47; *p* = 0.47) (Figure 2).

To adjust for differences in the recovery rate between the fairway and the green, the data were converted into a percentage recovery (adults vacuumed up divided by adults in total). The percentage recovery via vacuuming was significantly affected by the mowing height (F = 93.06; df = 1, 31; *p* < 0.001) but not by the vacuuming duration (F = 0.79; df = 1, 31; *p* = 0.38); both factors did not interact significantly (F = 0.16; df = 1, 31; *p* = 0.69). On the fairway, only 4 ± 1% and 5 ± 1% of the total recovered adults were collected by vacuuming for 15 and 30 s, respectively (Figure 2). On the green, 29 ± 5% and 33 ± 2% of the total recovered adults were collected by vacuuming for 15 and 30 s, respectively.

### 3.3. Effect of Water Volume and Detergent Concentration on ABW Adult Extraction Efficiency of Soap Flushing

In the 2019 experiment, the treatment (i.e., extraction methods that varied in the combination of water volume, detergent concentration, and the number of solution applications) had a significant effect on the total number of adults extracted after 20 min (F = 10.23; df = 4, 94; *p* < 0.001). For treatments that received two applications of 500 mL detergent solution, extraction increased with the detergent concentration, significantly higher at 0.8% (28.2 ± 3.7 adults) than 0.2% (16.2 ± 2.7) and intermediate at 0.4% (20.2 ± 2.7) (Figure 3). When the same detergent concentration (0.4%) was used, extraction increased with the soap solution volume (one application of 500 mL (9.0 ± 1.9) vs. one application of 1000 mL (15.7 ± 2.8)) and was higher when the same total amount of solution was applied in two portions of 500 mL (20.2 ± 2.7) vs. all at once (15.7 ± 2.8) (Figure 3). However, the application of 1000 mL at once always resulted in a significant run-off of the solution.

In the 2020 experiment, adult densities, and with that extraction numbers, were much lower than in 2019, but the observed trends were similar. The treatment (=extraction method) (F = 7.37; df = 4, 74; *p* < 0.001) had a significant effect on the total number of adults extracted after 20 min. As in 2019, the extraction efficiency increased with the detergent concentration of the solution. This was observed with two applications of 500 mL soap solution (0.8% (3.7 ± 0.5) vs. 0.4% (2.1 ± 0.4)) and one application of 500 mL (0.8% (2.3 ± 0.4) vs. 0.4% (0.9 ± 0.2)) (Figure 3). Extraction was similar when the same total amount of 0.4% solution was applied in two portions of 500 mL (2.1 ± 0.4) vs. 1000 mL at once (1.1 ± 0.4), but the application of 1000 mL at once always resulted in significant run-off. Unlike in 2019, extraction was similar with one application of 1000 mL of 0.4% soap solution (1.1 ± 0.4) and one application of 500 mL of 0.4% soap solution (0.9 ± 0.2) (Figure 3).

Overall, the highest extraction rate was observed in both years with two applications of 500 mL at a 0.8% detergent concentration. Moreover, in both years, the total extraction after 15 min and 20 min followed a similar pattern (Figure 3). Among the different treatments and across both years, the cumulative extraction efficiency relative to the total after 20 min varied more in the earlier sampling intervals (10–41% after 5 min, 36–72% after 10 min, and 71–90% after 15 min).

### 3.4. Effect of Time of Day on ABW Adult Recovery in Mower Clippings

The total number of adults recovered (i.e., adults in clippings plus soap extraction) was marginally higher in the afternoon samples (40.7 ± 3.7) than in the morning samples (37.4 ± 4.3) (F = 3.81; df = 1, 71; *p* = 0.055); the co-factor temperature had no significant effect (F = 2.69; df = 1, 71; *p* = 0.11). However, the percentage of adults collected in clippings (relative to the total number from clippings plus soap extraction) was significantly higher in the early afternoon samples (9.4 ± 1.0%) than in the morning samples (4.7 ± 0.7%) (F = 8.74; df = 1, 71; *p* < 0.01). The percentage of adults collected in clippings was also significantly affected by the co-factor temperature (F = 31.02; df = 1, 71; *p* < 0.001).

### 3.5. Effect of Time of Day on ABW Adult Recovery via Vacuuming on Fairway and Green

The total number of adults recovered was not affected by the time of day (F = 0.13; df = 1, 63; *p* = 0.72) but was higher in the fairway than the green (F = 268.88; df = 1, 63; *p* < 0.001) and higher for treatments that combined the vacuum and soap than for soap alone (F = 10.22; df = 1, 63; *p* < 0.01), with no interaction between factors (F ≤ 1.00; df = 1, 63; *p* ≥ 0.32) (Figure 4). Hence, recovery using soap alone was also not affected by the time of day in the fairway and the green (Figure 4). Adult numbers recovered by the vacuum alone were not significantly affected by the time of day (F = 0.05; df = 1, 31; *p* = 0.83) but were higher in the fairway than the green (F = 17.07; df = 1, 31; *p* < 0.001); there was no interaction between time of day and grass type (F = 0.05; df = 1, 31; *p* = 0.083). The percentage of adults recovered by the vacuum (relative to the total from the vacuum plus soap) was not affected by the time of day (F = 0.04; df = 1, 31; *p* = 0.85) but was higher in the green (5–6%) than the fairway (2%) (F = 4.33; df = 1, 31; *p* < 0.05); there was no interaction between the time of day and the grass type (F = 0.05; df = 1, 31; *p* = 0.83).

## 4. Discussion

Our findings provide important insights into the efficiency of sampling methods for adult ABWs and how they can be optimized. Overall, soap flushing is the most efficient and robust method and can provide precise population density information under varying field conditions. Vacuuming and, especially, clippings monitoring are less efficient and affected by mowing height, and clippings monitoring is also affected by temperature. However, other factors (speed, convenience) not tested here need to be considered as well in determining the usefulness of each method in IPM programs in golf courses.

Soap flushing has been used for a long time for sampling various surface-active insects in turfgrass, including cutworms, armyworms, sod webworms, mole crickets, and billbugs [1,10]. However, methodologies provided in textbooks or extension publications are generic, and no extraction efficiencies are known. Potter [10] recommends using 0.4% detergent (type not specified) applied once at 813 mL per 0.1 m2. Vittum [1] recommends a range of 0.19–0.75% detergent (lemon-scented preferred) applied once at 667–2667 mL per 0.1 m^2^. Tashiro et al. [13], specifically for the grass webworm, *Herpetogramma licarsisalis* Walker, found the most effective detergent concentration and volume to be 0.25% and 1750 mL per 0.1 m^2^ applied once.

In our study, the soap flushing extraction efficiency of adult ABWs increased with detergent concentration, whether the solution was applied in one (1000 mL at 0 min) or two portions (500 mL at 0 and 5 min). The effect of the solution volume (500 mL or 1000 mL at 0 min) was less clear. In the 2019 experiment, almost twice as many adults were recovered with one application of 1000 mL than with one application of 500 mL, but there was no significant effect in 2020. Overall, the application of 0.8% solution in two portions of 500 mL applied at 0 min and 5 min (total of 1075 mL per 0.1 m^2^) was clearly the most efficient method and will be referred to henceforth as the optimal method. We did not test higher detergent concentrations than 0.8% as we expected such a high concentration to be detrimental to the adults, thereby reducing the efficiency, and potentially detrimental to the turfgrass. The absolute extraction efficiency of the optimal method could not be determined as it was only tested using natural ABW populations, but in that experiment, 1.6 times more adults were recovered with 0.8% detergent than with 0.4%. Since in the two experiments using marked weevils 75–80% of the released adults were recovered using two applications with 0.4% detergent, the optimal method (0.8%) can be expected to extract close to 100% of adults.

The soap flushing extraction efficiency was not significantly affected by the temperature or time of day. In the experiment examining the effect of the temperature on adult extraction with vacuuming, there was no difference between the number of adults recovered with soap alone in the mornings (12.2–15 °C) or in the early afternoon (20.0–20.6 °C). Even with greater temperature extremes, as in the experiment examining the effect of the temperature on adult extraction in mower clippings (6.7–26.1 °C), there was no effect from the temperature or time of day on soap extraction. It has to be noted that while the temperature and time of day are obviously related factors (with temperatures in our experiments always higher in the morning than in the early afternoon), there are also other factors that may impact recovery in the morning vs. the early afternoon. Leaf wetness tends to be high in the morning because of dew. The relative humidity is likely to be higher in the morning as well, although, at least within the turf canopy and, to a lesser extent, directly on the canopy, i.e., the areas where adult ABWs are active, the relative humidity will tend to be high as the short-mown areas on golf courses are irrigated more frequently so that the soil and thatch under the canopy will usually stay moist. Insolation is obviously higher in the early afternoon than in the morning, but based on empirical observations, adults can be active at any time of the day and even during the night, at least during warm nights [1; A.M.K., personal observation].

Two disadvantages of the soap flushing method are issues with run-off from the solution and the need for the repeated collection of the adults over up to 20 min. Run-off from the detergent occurred on fairways and particularly on greens where even 500 mL had to be applied slowly. The application of 1000 mL at once was impossible without significant run-off even on fairways. On surfaces with even a small gradient, the run-off would be even more problematic. Higher solution volumes, as suggested elsewhere [1,13], may be feasible in taller and less dense grass and on very sandy soils but not under the conditions our study was conducted. The other issue, sampling time, could be reduced by shortening the observation period. Thus, of the total number of adults recovered in each sample (i.e., cumulatively after 20 min) using the optimal method, 20.9 ± 0.6% (mean ± SEM) were recovered after 5 min, 54.7 ± 0.8% were recovered after 10 min, and 88.6 ± 0.4% were recovered after 15 min. Hence, for the sake of saving time, sampling could be concluded after 15 min or even 10 min with a fairly good representation of the total population and a very small variation in the recovery rate.

An advantage of the soap flushing method is that several samples can be taken in parallel. In our experience, five samples of 30.5 cm × 30.5 cm can be easily conducted in parallel by one person, allowing for a good representation of a larger monitoring area. If the adults are collected every 5 min for 20 min, the total time taken to sample an area is about 30 min, including setting up, preparing the soap solution, collecting the adults, rinsing the plots with water, and collecting all the material when complete. As discussed above, this can be reduced to about 20 min if adults are only collected for 10 min. Sampling five samples requires 5 L of water for the soap solution and 5 L for rinsing the plots. The water can be transported in large collapsible plastic jugs, e.g., 5-gallon (19.12 L) jugs, of which many fit on the back of a typical golf course utility vehicle, allowing for the sampling of many areas.

Vacuuming has been adopted for sampling various surface-active insects, such as herbivores (e.g., beetles, planthoppers, true bugs, etc.), predators, and parasitoid wasps, in vegetation [14,15,16,17], including ABWs in turfgrass [4,18]. In our study, vacuuming was more effective in recovering adult ABWs on greens than on fairways. In the experiment with released adults and at high temperatures (21–27 °C), one pass of vacuuming recovered 29% of adults from the green but only 4% from the fairway. A second pass did not significantly increase recovery and thus does not seem worth the extra time it would add to monitoring. The same effect was observed in the experiment with natural populations, albeit the difference between recovery at the green (4–5%) and the fairway (2%) was much smaller than in the experiment with released adults. Similarly, Brook et al. [14] also reported that increasing vegetation height negatively affected the capture efficiency of suction sampling in grasslands, consistent with the findings of Hossain et al. [15]. The temperature did not have a significant effect on recovery in our study. It is possible that a lower portion of the marked adults settled deeper into the turf between the grass plants, where they would be harder to dislodge from the turf. The total recovery of marked adults (i.e., including soaping) was significantly lower from greens than from fairways, probably because adults can move faster over the surface of greens than fairways, and thus, some might have walked outside of the green plots before the mowing started. Indeed, in the fairway, the adults tended to be recovered closer to the release area within the soap-flushed samples, whereas they tended to be more dispersed in the green. The same observation was made in the experiment on mower clippings with marked adults.

Vacuuming can be performed fairly easily with little equipment to carry around, but the extraction efficiency, at least on fairways, is low (2–4% with one pass). To sample an area adequately, at least three passes of at least 10 sec each should be conducted based on our empirical observations. Executing a pass would take about 1 min followed by anywhere from 1 min to about 4 min to go through the sample, depending on how much debris is picked up by the vacuum. On greens, there is typically very little debris as the clippings are collected with each daily mowing, and the green’s surface is cleared of debris regularly to allow for unobstructed ball rolling. On fairways, on the other hand, clippings are not collected on many golf courses, and much debris can collect on their surfaces. As samples cannot be taken simultaneously by one person, three 10 sec samples can be taken accordingly between about 6 min and 15 min.

The extraction of adult ABWs in mower clippings was clearly affected by the mowing height and temperature. At high temperatures (18–25 °C), on fairways only 0.5% of recovered adults (marked) were found in the clippings, whereas on greens 15% (marked; without brush) or 10% (natural population) of adults were found in the clipping. Recovery of adults in mower clippings decreased with temperature from around 10% above 20 °C to around 6% at 15 °C to 3–4% at ≤10 °C. Hence, this monitoring method only works on greens and would work better in warmer temperatures, i.e., generally later in the day. However, since greens need to be mowed before the golfers reach them, this method may not be reliable enough for monitoring the overwintered adults, as morning temperatures during the period, when this needs to be performed in spring, are generally still low. Furthermore, while monitoring adults in greens in mower clippings could be easily incorporated into the typical mowing practices of a golf course, going through mower baskets full of clippings would also take a very long time, so a subsampling method would need to be developed and quantified. As an aside, daily mowing can significantly reduce adult populations over time in greens, particularly at the lower end of mowing heights for greens [12]. Adding a brush in front of the mower baskets could further increase adult removal, as it increased adult recovery from 15% without a brush to 24% with a brush in our experiment. However, with around 70% of the adults recovered in the clippings being unharmed, it is crucial that the clippings are disposed of in a way that will prevent these surviving adults from re-infesting susceptible turfgrass areas.

Overall, our observations show that soap flushing should be the preferred method for monitoring adult ABWs. Vacuuming might be a viable alternative on greens, where the transportation of sufficient amounts of water may be problematic for soap flushing and if time constraints are more important than precision. Monitoring adults in clippings on greens may be used to observe general trends in the population but does not appear feasible for more precise observations. The optimal soap flushing method determined in this study should help superintendents more precisely monitor adult ABW populations and minimize insecticide applications for ABW management in time and space, thereby decreasing the risk of insecticide resistance development.

## Figures and Tables

**Figure 1 insects-14-00509-f001:**
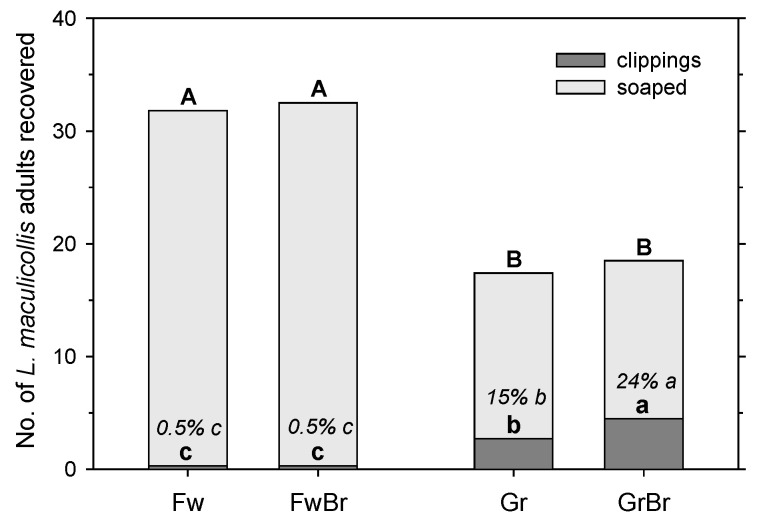
Recovery of marked *Listronotus maculicollis* adults from areas mown at fairway height (Fw) and green height (Gr) using a mower with (Br) or without a brush attached to the front of the mower basket followed by soap extraction. Letters indicate significant differences between the number of adults recovered from mower clippings (lowercase) and from clippings and ensuing soap extraction combined (capitals) (*p* < 0.05). Percentages within bars are percentages of adult recovery in clippings relative to total recovery; lowercase italicized letters indicate significant differences between these means (*p* < 0.05).

**Figure 2 insects-14-00509-f002:**
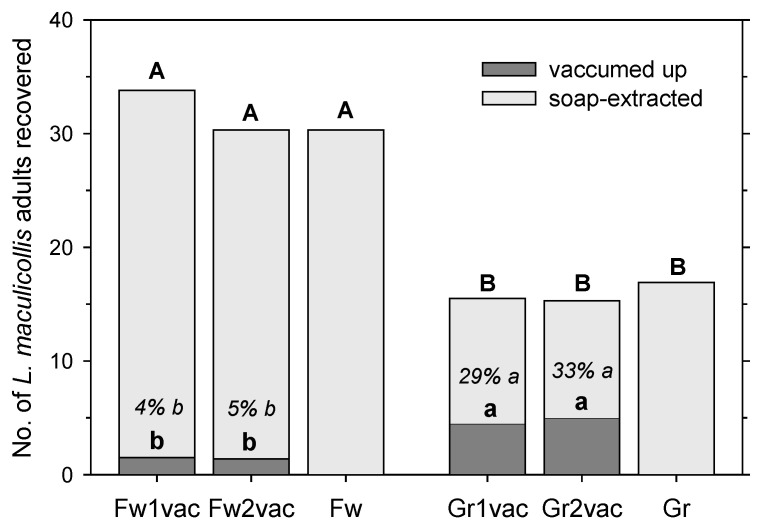
Recovery of marked *Listronotus maculicollis* adults from areas mown at fairway height (Fw) and green height (Gr) by no, one (1vac), or two (2vac) passages with a vacuum followed by soap extraction. Letters indicate significant differences between the number of adults recovered by one or two vacuum passages (lowercase) and via vacuuming and ensuing soap extraction combined (capitals) (*p* < 0.05). Percentages within bars are percentages of recovery in clippings relative to total recovery; lowercase italicized letters indicate significant differences between these means (*p* < 0.05).

**Figure 3 insects-14-00509-f003:**
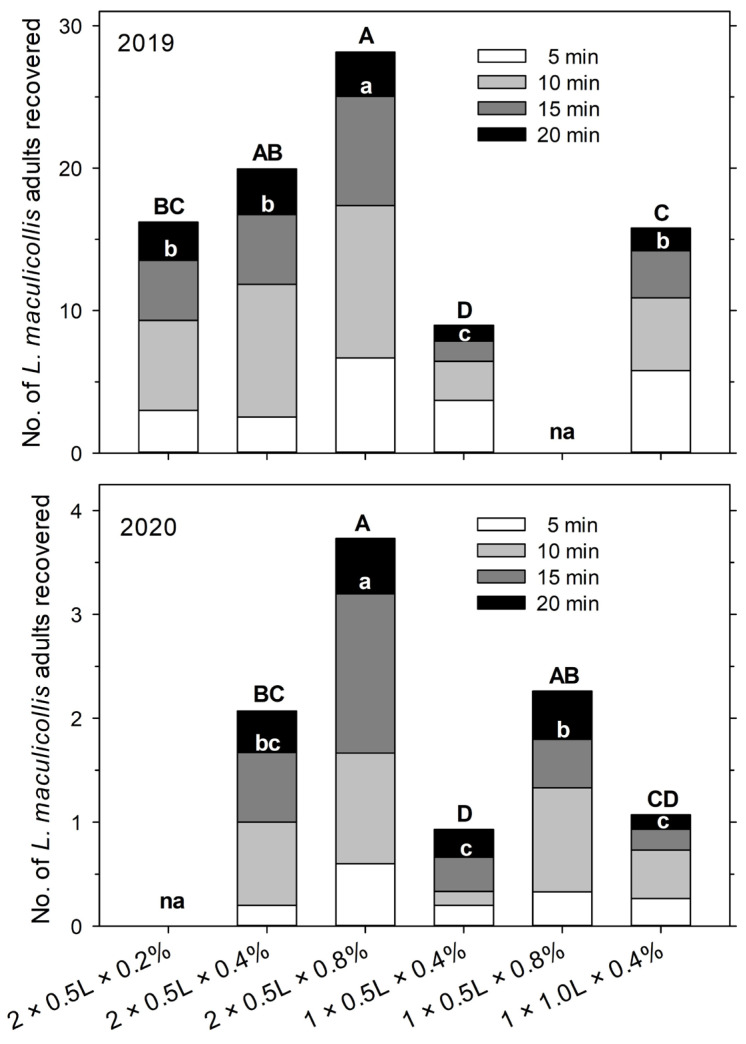
Recovery of *Listronotus maculicollis* adults in experiments conducted in 2019 and 2020 from areas mown at fairway height with soap solution flushes at different application frequencies (1× at 0 min or 2× at 0 and at 5 min), water volumes (0.5 L or 1.0 L per application), and detergent concentrations (0.2%, 0.4%, or 0.8%). Adults were collected at around 5, 10, 15, and 20 min after the first application of the soap solution. NA indicates that the treatment was not included in that year’s experiment. Upper- and lowercase letters indicate significant differences between treatments in the total number of adults recovered from 5 to 20 min and 5 to 15 min, respectively (*p* < 0.05).

**Figure 4 insects-14-00509-f004:**
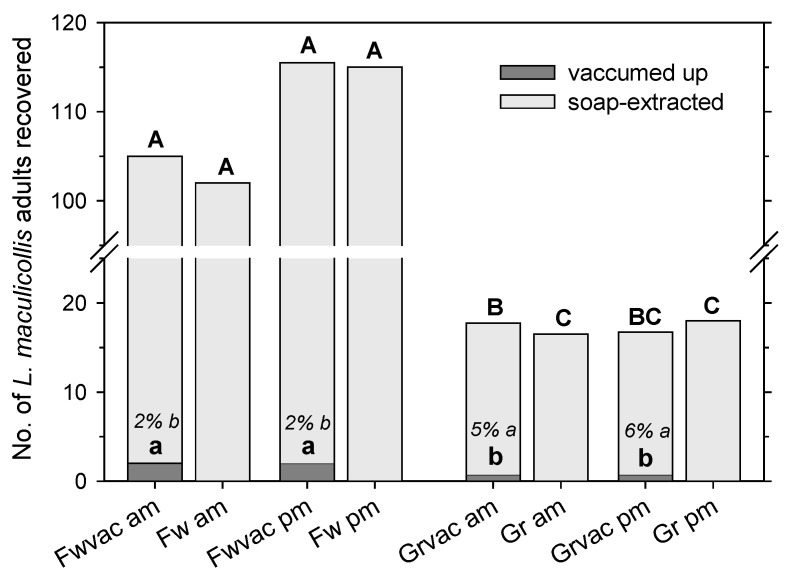
Recovery of *Listronotus maculicollis* adults from areas mown at fairway height (Fw) and greens height (Gr) by no or one (vac) passage with a vacuum followed by soap extraction. Experiments were conducted in the morning (a.m.; 12.2–15.0 °C) or early afternoon (p.m.; 20.0–20.6 °C). Letters indicate significant differences between the number of adults recovered by vacuuming alone (lowercase) and the total number by vacuuming and ensuing soap extraction combined (capitals) (*p* < 0.05). Percentages within bars are percentages of recovery via vacuuming relative to total recovery; lowercase italicized letters indicate significant differences between these means (*p* < 0.05).

## Data Availability

The data presented in this study are available upon request from the corresponding author.

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
