# Peer review of "Optimizing Sampling Technique Parameters for Increased Precision and Practicality in Annual Bluegrass Weevil Population Monitoring"

_insects, 2023, doi:10.3390/insects14060509_

Round 1

Reviewer 1 Report

This study provides a sampling technique for Listronotus maculicollis in short-mown turfgrass. Although there are already some works on sampling this insect in similar environments, the authors evaluate in this work a series of parameters that had not been previously optimized and arrive at a recommendation based on consistent experimental results.

I consider that the work meets the conditions to be published, but some details should be reviewed by the authors beforehand: 

In the results section: Values after ± are standard error of mean?

Line 319 (interpretation of figure 3): "...was similar when the same total amount of solution was applied in two portions of 500 mL (20.2 ± 2.7) vs all at once (15.7 ± 2.8) (Figure 3)." You say that was similar, but averages are in different groups (AB and C). 

Line 359 (interpretation of figure 4): "and higher in treatments that combined vacuum and soap than for soap alone": This doesn't seem right. In no case is it seen that the treatment with vacuum has a mean in a different capital letter group than the mean of the respective treatment without vacuum.

Line 36: "Adult numbers recovered by vacuum alone were not time of day...": This sentence is unclear. 

Lines 386-387: "Soap flushing has already been used for a long time for sampling various surface- 386 active insects in turfgrass including cutworms, armyworms, sod webworms, mole crickets  387 and billbugs": Include the corresponding references for each case. 

Comments are also included in the attached pdf

The quality of English is appropriate. A review is recommended for minor changes.

Author Response

Reviewer #1:

-In the results section: Values after ± are standard error of mean?

AU:  The following sentence was added at end of section 2.7. (Statistical Analysis): “Means presented in the text are followed by ± SEM (standard error of the mean).”

-L245:  analysis of variance -> factorial analysis of variance

AU:  “factorial” was added.

-Line 319 (interpretation of figure 3): "...was similar when the same total amount of solution was applied in two portions of 500 mL (20.2 ± 2.7) vs all at once (15.7 ± 2.8) (Figure 3)." You say that was similar, but averages are in different groups (AB and C).

AU:  was changed to “and was higher”.

-Line 359 (interpretation of figure 4): "and higher in treatments that combined vacuum and soap than for soap alone": This doesn't seem right. In no case is it seen that the treatment with vacuum has a mean in a different capital letter group than the mean of the respective treatment without vacuum.

AU:  The statement is for all data analyzed together which, absent of a significant interaction between factors, were significantly higher for vacuum+soap.  And there was actually one case where there was a significant difference within same treatment:  Grvac am (B) > Gr am (C).

-Line 36: "Adult numbers recovered by vacuum alone were not time of day...": This sentence is unclear.

AU:  changed to “Adult numbers recovered by vacuum alone were not significantly affected by time of day...

-Lines 386-387: "Soap flushing has already been used for a long time for sampling various surface active insects in turfgrass including cutworms, armyworms, sod webworms, mole crickets and billbugs": Include the corresponding references for each case.

AU:  The method is recommended for surface active turfgrass insects in the various turf entomology textbooks and numerous extension publications.  We could not find any more specific references for the various pest for which it is recommended in the textbooks.  And these textbooks are mentioned in the following sentence.  Nonetheless, we added the references to these textbooks also at the end of the first sentence ([1, 10]).

-L420:  tends to high -> tends to be high

AU: “be” was added.

Author Response

Reviewer #2:

-It would be useful to know what sort of numbers/density of weevils should prompt action by greenkeepers, say when to apply pesticides. 

AU:  Some info was added about damage thresholds in the 3rd paragraph of the introduction. 

-Were the numbers of weevils added to plots in the realm of numbers found naturally, or as it seems, were they higher.  How many would be required to have damaging effects to greens and fairways.

AU:  The densities added in the form of marked weevils were well within the realm of natural populations, albeit on the higher side, as can be seen in the high adult numbers in the soap flushing experiment in 2019 (up to 30 per 0.093 m2).

-Also, is time of year for monitoring a consideration. The Introduction notes that there can be 2-3 generations a year, so timing of monitoring might be important. Other specific comments noted below.

AU: The papers is about the sampling methods, not about ABW in general or damage thresholds etc.  Hence, for the sake of brevity and flow, we don’t want to add too much additional info to the introduction or discussion. Monitoring is obviously important during the entire season.  But managing, and with that also monitoring, ABW is generally considered most important in spring so that the populations cannot build up too much during the season.

Line 42: What is meant by ‘high profile’ – prestigious or elevated.

AU:  “high profile” is supposed to mean areas that are under particular scrutiny because they are expected to be in very good shape for various reasons like importance to the game, most likely to be noticed by the golfers, etc.  Not sure how to say that differently without using this many words.  Have used it numerous times before, and seen other people use it for same purpose, without issues.

Line 64: delete ‘enough’

AU: deleted

Lines 68-72: need to acknowledge, however, that these methods are more accurate for population estimation.

AU: Does the sentence before not clearly do that?  “Larval monitoring has a greater predictive power since the larvae are the stage causing significant damage and adult densities only indirectly predict larval densities.”

Line 75: assume by ‘inverted’ you mean a leaf blower set on suction?

AU:  Correct.  We changed to “leaf blower with inverted air flow” to clarify.

Line 87-88: hard to envisage mowing in a greenhouse pot experiment, so assume this means clipping to simulate mowing?

AU:  They actually use something pretty close to a mower:  pots exactly arrange on a bench and a mower-like machine running at an exactly set height over the pots.  I wish we had something like that, even just to mow the grass in the greenhouse rather than having to use scissors!

Line 125: remove space between ‘fairway’ and ‘s’.

AU:  space removed.

Line 135-136: this is unclear to me – was this the centre of the 183x61cm plot? If so just indicate that – but 10cm widthwise does not suggest this.

AU:  Sorry, we thought this was clear.  The plots are 183 cm long and we released adults in the center 30.5 cm of that which is 76.25 cm from both ends.  The plots are 61 cm wide, and we released adults in the center 10 cm which is 25.5 cm from both sides.  Changed it to “Forty color-marked ABW adults were released in a 30.5 cm long and 10 cm wide area in the center of the plot (i.e., at least 76.2 cm and 25.5 cm from the plots borders length- and widthwise, respectively) and allowed to disperse for 40 min.”  We hope this is clearer now.

Line 148: is this collection by hand and transferring to cups? I see from 2.4 this is explained – perhaps clarify earlier.

AU: We added “and transferring them with a soft forceps into”…

Line 149: two litres?

AU: Changed.

Line 152: Add s to ‘Adult’

AU: done.

Line 163: unclear what is meant by ‘two experimental runs’ – so two replicates of the 12 plots?  Reading further down I see that the two ‘runs’ were experiments repeated on different dates.  Perhaps clarify this earlier.

AU:  We deleted the “runs’ part as not necessary here.  At the bottom of section, we clearly state that the experiment was run twice on two different dates.

Line 192-206: this would be much clearer in a table.

AU: We tried a table, but it did not seem clearer and took up more space.  Instead, we added a reference to the corresponding figure which might help understand better.

Line 228: define RCB

AU: wrote out “randomized complete block”.

Statistical Analysis: needs a sentence here to explain how the data from marked weevils were used.

AU: Does the reviewer mean that we calculated percent extraction based on the number of marked weevils released?  That was our original plan.  But because the total numbers recovered (including the soaping) were much lower on the greens and we did observe some adults take flight on the greens, we did not end up doing that.  Instead, we just used the numbers recovered.  Percentage recovery rather was calculated based on the total recovered including soaping as the latter was so much more effective at recovering the adults than the mowing and vacuuming.  We added a sentence at the beginning of the section on Statistical Analysis to clarify this.

Fig. 1: use ‘soap-extracted’ for legend as in Fig. 2 – consistency and more descriptive

AU: was added.

Line 331: are the lower numbers in 202 because no marked weevils were added – just naturally occurring weevils. Be good to make this clearer in the methods.

AU:  Both experiments (2019 and 2020) were conducted with natural populations as indicated in the material and methods (4th line in section 2.4).

Line 380: I would question the statement that the soap extraction gives ‘precise population density information’. The data have not been presented in these terms. In some cases, known numbers of weevils were added to plots, but actual numbers still in the sampled areas was not known, and the natural population densities were unknown. Also if the soap solution could be confined to known areas with a sunken quadrat of known area this would be more precise given you have determined the proportion of marked weevils recovered of those released. Perhaps I have misunderstood – if so, further clarification would be good.

AU: See discussion at end of third paragraph.  In experiment with marked weevil 75-80% of marked weevils were recovered.  We would call that quite precise already.  And that was achieved with the 0.4% detergent solution which, as shown in the experiment with natural populations, is significantly less effective than the 0.8% solution.  Hence, the optimal method with 0.8% can be expected to be even more efficient and hence even more precise.

            As to using a sunken quadrat: that would leave clear marks in the grass which on a green would be totally unacceptable and would not be acceptable on a lower cut higher quality fairway either.  We don’t fully understand the comment, though: we did determine populations within a defined area (30.5 x 30.5 cm in the soaping experiment).  To assuage the reviewer's concerns, we added when we first describe the soaping method in the last paragraph of section 2.2 that “Adults were collected from within that defined area only as done in all following experiments.”

Line 479: On, not ‘One

AU: corrected

Round 2

Reviewer 1 Report

The authors followed the recommendations and provided all the necessary clarifications. I consider the work to be publishable in its current state.